# Methodologies of Learning Served by Virtual Reality: A Case Study in Urban Interventions

**Monica V. Sanchez-Sepulveda** [1,*] **, Ricardo Torres-Kompen** [1] **, David Fonseca** [1] **and Jordi Franquesa-Sanchez** [2]

[1]  Group of Research on Technology Enhanced Learning (GRETEL), La Salle, Ramon Llull University, 08022 Barcelona, Spain; ricardo.torres@salle.url.edu (R.T.-K.); david.fonseca@salle.url.edu (D.F.)

[2]  Barcelona School of Architecture, Polytechnic University of Catalunya, 08028 Barcelona, Spain; jordi.franquesa@upc.edu

*  Correspondence: monica.sanchez@salle.url.edu; Tel.: +34-602-090-2052

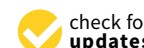

**Featured Application: Methods for educational purposes, such as courses related to Urban Design in Architecture Schools.**

**Abstract:** A computer-simulated reality and the human-machine interactions facilitated by computer technology and wearable computers may be used as an educational methodology that transforms the way students deal with information. This turns the learning process into a more participative and active process, which fits both the practical part of subjects and the learner's profile, as students nowadays are more technology-savvy and familiar with current technological advances. This methodology is being used in architectural and urbanism degrees to support the design process and to help students visualize design alternatives in the context of existing environments. This paper proposes the use of virtual reality (VR) as a resource in the teaching of courses that focus on the design of urban spaces. A group of users—composed of architecture students and professionals related to the architecture field—participated in an immersing VR experience and had the opportunity to interact with the space that was being redesigned. Later, a quantitative tool was used in order to evaluate the effectiveness of virtual systems in the design of urban environments. The survey was designed using as a reference the competences required in the urbanism courses; this allowed the authors to identify positive and negative aspects in an objective way. The results prove that VR helps to expand digital abilities in complex representation and helps users in the evaluation and decision-making processes involved in the design of urban spaces.

**Keywords:** information and communication technologies; higher education; urban design; virtual reality; learning methodologies; architecture schools; teaching/learning strategies; interactive learning environments

## 1. Introduction

Any research in the educational field should start by considering how students learn. Ambrose [1] states that learning is a process and not a product, but since this process takes place in the mind, it can only be assumed that it has taken place by analyzing changes in the learners. Whenever a professor designs a class, the first issue that must be considered is the audience. The nature of students—their academic preparation, aspirations, and cognitive development—affects the professor's elections on what and how to teach. The focus should not be on the teaching of the specific contents of the subject (such as physics, Spanish, mathematics, art, and so on), but on the students as an audience [2]; this requires an understanding of how the human mind learns. There are certain kinds of delivery that are

more effective in terms of communication than others, meaning that they make it easier for people to pay attention, remember and grasp concepts, and process information and knowledge [2].

Centuries of teaching at University level have been characterized by an audience of students taking notes from a professor delivering authoritative lectures, to later test students on their knowledge and assign grades [3]. However, over the past thirty years, research on theories of learning and cognitive development and students' academic success has confirmed that teaching that emphasizes active learning and collaborative activities, and that promotes intellectual engagement on students, is more efficient. Professors interact with students in ways that allow the latter to obtain new information, practice new competences, and rethink and expand on what they already know. Other relevant concepts and ideas provided by previous studies [2], that will also be taken into account in this study, focus on several characteristics of human beings. According to these studies, they are as follows:

- Are innate learners, capable of remembering and absorbing uncountable details about objects and other people [4,5]?
- People Learn by connecting new information to what they already know [4,6].
- People learn what they consider relevant to their lives [7].
- People learn informally by building knowledge in social groups [8], but also learn individually and in one-on-one situations [5].
- People learn when they are motivated to do so by receiving encouragement from other people in their lives [9].
- People do not learn satisfactorily when their main learning environment is professor-centered, and it requires passively listening while the professor talks. Human beings cannot pay attention for long when their brain is in an inactive state [5,7,10–14].
- People learn more when they obtain new material several times by using diverse methods, which require the use of different parts of their brain [15–17].
- People learn when they actively examine their learning and performance [4].
- People learn less by going through the material and more from being examined by others or themselves on it, as it implicates more cognitive processing and requires them to practice retrieving information [18–20].
- People learn more when the material helps stimulate emotions and not just intellectual or physical involvement [21–23].

For most people, part of the value of a career in academia is the chance to help learners in their path to becoming future professionals in the field, and share their enthusiasm with others; for this reason, it can be discouraging for them to look into a classroom and see disengaged students. This lack of engagement and motivation happens mostly when it is hard for the learners to connect what they are learning to existing knowledge; when they are learning passively and individually; when the information is explained just one time or in only one way, making it more difficult to understand its relevance; and when the material does not evoke emotions and does not produce an action-reaction experience. For students to understand the concepts being presented to them, it is crucial that they are engaged. The more the student is engaged in academic work, the greater the level of knowledge achievement and general cognitive expansion.

Information and communications technologies (ICTs) have transformed our society and, by extension, education [24]. The methods in which we communicate have been continuously adjusting to include characteristics such as interconnection, interaction and mobility. Several studies [25] show the opportunities offered by these emergent technologies, as they allow for a new type of reality where physical and digital environments, media and interactions are intermingled through our everyday lives. The use of interactive virtual systems, in particular, has begun to be used in professional and educational sectors, as it allows users to be close to the space recreated while letting a fast flow of the changes of the model made in real time [26].

Recent studies [27] focus on the adaptation of contents and their application using ICTs in the fields of architecture and urban design, focusing on the student, and their satisfaction and motivation. From an academic perspective, ICTs enhance the acquisition of spatial competences to study the visual impact of urban or architectural projects. Particularly, in architecture and urban design courses, it is necessary to evaluate whether a design is appropriate before being built, leading educators to reconsider how students represent the designs and learn to make this evaluation. Thus, it is important that students develop skills in various representation technologies, and can integrate the latest technologies in their design process, with the aim of better communicating their proposals, and to facilitate analytical thought on the spaces they design [28].

Throughout the history of architecture education, understanding and visualizing 3D spaces has been usually done by means of drawings and physical models instead of 3D models and virtual visualizations [29]. The use of these new methods is emerging because of generational change and the continuous development and improvement of technology [30]. New technologies are transforming the way these processes are carried out. The world that surrounds us is now becoming progressively digital, in particular for newer generations that are technologically savvy and familiar with the use of mobile devices and cloud computing services [31], and the integration of this new approach and paradigm in the specific context of the education and professional practice of urbanism is crucial.

In education, mixed reality (MR) is a new approach that transforms the way students deal with information. MR technologies, namely virtual reality (VR) and augmented reality (AR), are used in architectural and urbanism research and practice to support the design process, to visualize design alternatives set in existing built environments, and to assess people's reaction to their living environment.

Virtual reality (VR) is a combination of technologies used to visualize and provide interaction with a virtual environment. The variety of settings in which VR could be used to represent make it largely relevant to many areas in education. An important characteristic of VR is that it allows for multi-sensory interaction with the space being visualized. The combination of multi-sensory interactivity makes VR ideally suitable for efficient learning, as it benefits from the advantages provided by active learning through experiences, which is the main reason why we choose this system for this case study [32]. Virtual reality is broadly used in the industry and is starting to be more affordable for users. AR and VR share features like interaction, navigation and immersion [33]; AR may be defined as a VR variation in which the user can see virtual objects mixed or superimposed upon the real world. In contrast to VR, AR does not replace the real environment; rather, it uses the real environment as a background.

One of the main motivations of university students is to be well prepared for their professional life and so they expect more courses to involve practical applications during their academic studies. Visual communication skills are connected to the competences required for professional practice. Architects should be able to choose a suitable representation medium, such as traditional graphics and digital technology tools, in order to communicate fundamental formal elements during each phase of the programming and design process.

## 2. Methods and Technologies

This case focuses on the subject Computer Tools II at the La Salle Architecture School (Ramon Llull University) during the academic year 2018–2019; in this course, students are introduced to emerging technologies such as augmented and virtual reality. The course is focused on using videogame technology for architecture representation [34], taking advantage of improvements in real-time rendering to produce interactive content. The students participated in an educational experience that sits at the intersection of architectural representation and urban design. The case study proposed to the students was part of an urban project promoted by the Barcelona Metropolitan Council, which aims to generate spaces that are designed to meet the needs of users. It takes into consideration that are spaces pleasant with vegetation, have dynamic uses, spaces for children's games, urban gardens, lighting, recreational and cultural activities, among others [35].

The case consisted in the re-urbanization of the area of Plaça (square) Baró, in Santa Coloma de Gramanet, Barcelona, according to the needs of the neighbors, which had been previously detected [35]. The course was split in two parts: first, the students were divided into groups of two students, and each group was assigned to work on a part of the urban environment of the square. During the following weeks, the groups modeled and textured the proposals of their respective sections, following simple guidelines regarding aspects such as maximum building size. At the end of this process, all the models were consolidated into a single environment, shared by all groups (Figure 1).

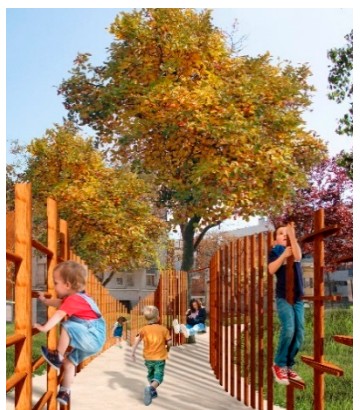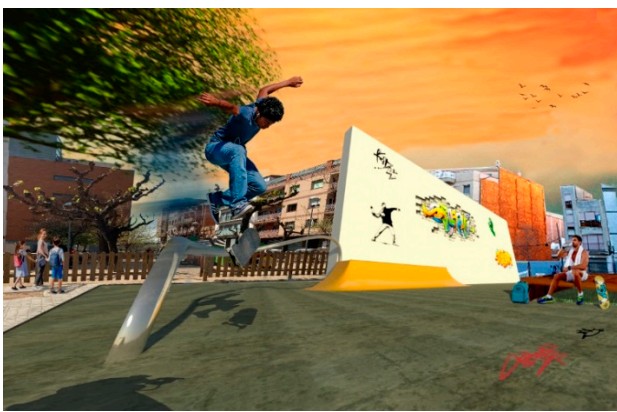

**Figure 1.** Examples of two students' proposals for their assigned section in Plaça Baró. One is a playground for children and the second one is a skate park for the youth.

The students work on these spaces to visualize them in real time, using Unreal Engine. The idea is that now neighbors and the city council are able to visualize the scale, the textures, the lights and shadows, amongst other elements, in the context of the needs and uses of citizens. Virtual reality allowed participants to see in an immersive way the changes and actions that happen in the environment in real time. For example, in the design of specific lighting—to be able to see the change from daylight to nightlight in a space, in a dynamic and realistic way. This powerful rendering engine allows for the calculation of lighting, and the user can (from a first-person perspective) design the lighting of an urban environment, try it on any section of the street and see how it is affected by the color, intensity or type of light being used [34] (Figure 2).

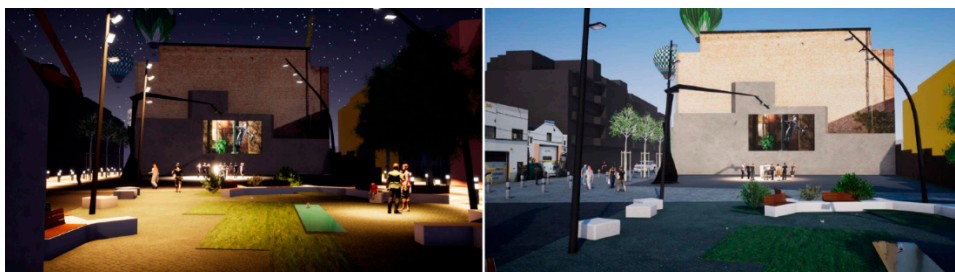

**Figure 2.** Dynamic view of night and day light. Work made by the students of La Salle—Ramon Llull University (URL).

This virtual three-dimensional scenario becomes an environment that users can interact with, in order to recreate new spaces. These spaces are meant to show maximum realism, including materials, textures, movements, and even sounds of the environment. Using VR glasses, the users experimented and shaped the urban public space. The VR let users understand in an immersive way how their actions and changes affect the environment in real-time (Figure 3). For example, having the capacity to be in continuous interaction with the open space while moving and rotating objects.

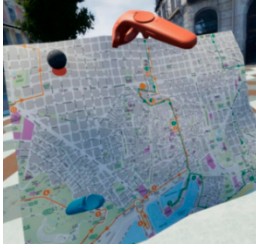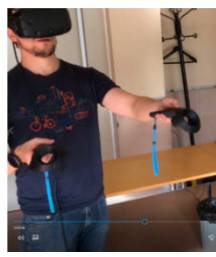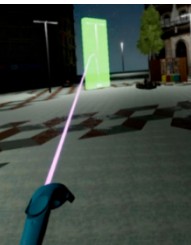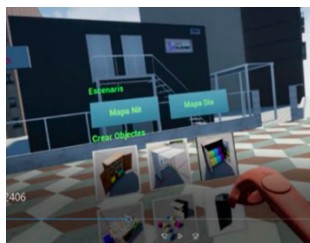

**Figure 3.** Examples of user's interaction with space and its effects. Two-hand joystick with different options: map to indicate your location in the site, grabbing objects to move or rotate and a catalog with urban furniture.

To validate that the virtual gaming application in the teaching of urban design projects enhances the spatial perception and urban competences, thanks to the immersive visual technologies, we have identified four main objectives that will help in detecting the essential elements that may help promote initiatives in both urban transformation and designing processes [36]:

- Assess the incorporation of immersive ICTs and gamification in the educational process (specifically in the urban project).
- Assess the motivation and usability of the gamification platform by the users.
- Study and establish links between the profiles of the users and the results of the surveys.
- Determine the relationship between satisfaction, motivation and user experience.

In architecture schools, urban design implies a dialogue between buildings aesthetics, scales and strategy, making possible the construction of the land and the city as a whole. This requires the architect to have a training in urbanism [37]. The competences (Table 1) required in urbanism courses in Spain are based on the White Book (Libro Blanco) that contains the design principles of studies and practical cases to be used in the design of a degree adapted to the European Higher Education Area (EHEA). These competences may be used as starting points for the creation of the survey and the assessment of the effectiveness of virtual systems in urbanism.

**Table 1.** Competences in the curriculum for urbanism of the School of Architecture, La Salle—URL that relates to each survey statement.

| Survey Statement # | Urbanism Competence |
|:---:|:---:|
| 1, 2 y 6 | Ability to comprehend the relations among people and buildings, between buildings and their surroundings, and buildings and spaces among them based on human scale and needs |
| 3 | Capability of making decisions (in projects, construction systems, organization, etc.) |
| 4 | Capability to communicate ideas, information, problems and solutions to a specialized and non-specialized public |
| 3 | Capability of acquiring self-critical capacity |
| 5 | Aptitude or ability to apply the basic formal, functional and technical principles to the conception and design of buildings and urban complexes, defining their general characteristics and benefits to be achieved |
| 7 | Aptitude or ability to develop building programs, considering the requirements of customers and users, analyzing precedents and location conditions, applying standards and establishing dimensions and relationships of spaces and equipment |
| 2 | Understanding the relationships between human behavior, the natural or artificial environment and objects, according to human requirements and scale |

To analyze and evaluate the advantages and disadvantages of virtual systems in the process of developing an urban and architecture project we surveyed architecture students of La Salle—Ramon

Llull University (URL) and professionals in the construction field at Construmat International Construction Fair, on their experience of using virtual systems during the design of urban environments. We asked users to wear VR glasses to evaluate how this method can help in designing urban spaces in our city, Barcelona (Figure 4) after having gone through the interactive virtual experience; this approach is similar to what has been done in previous experiments in the architecture educational framework [38,39].

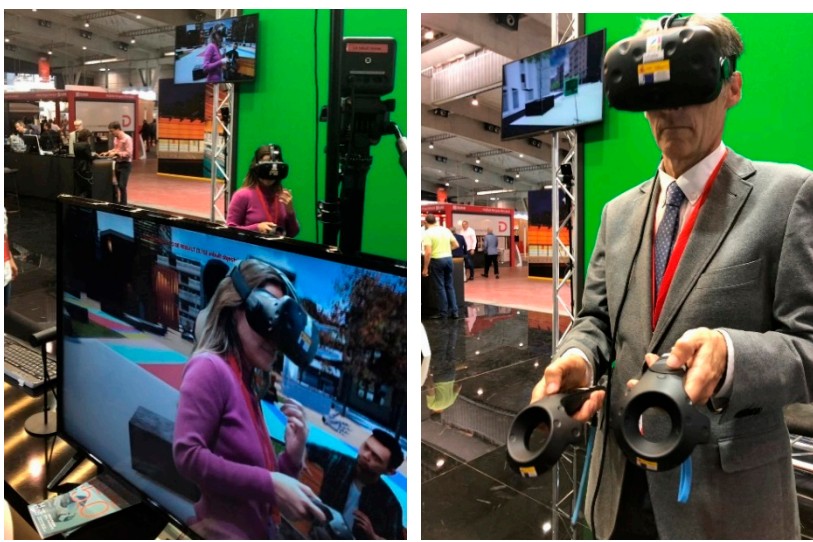

**Figure 4.** Participation of professionals in the Construmat International Construction Fair.

Some examples of quantitative methods used in scientific research are profile tests, satisfaction surveys and usability tests; if it is possible to work with big enough samples (of minimum 30–50 points), quantitative information can be collected, and the results can be analyzed and compared with the purpose of finding statistical differences [40]. The classic tool applied in similar scenarios is the survey, which is usually aimed at quantifying the effectiveness of a system, the users' opinion in general, and their level of satisfaction with the proposed method. The value of research lays not so much on the epistemology of the method, but on its efficiency [41], as quantitative methods are considered objective [42,43] and require deduction to interpret results (Figure 5).

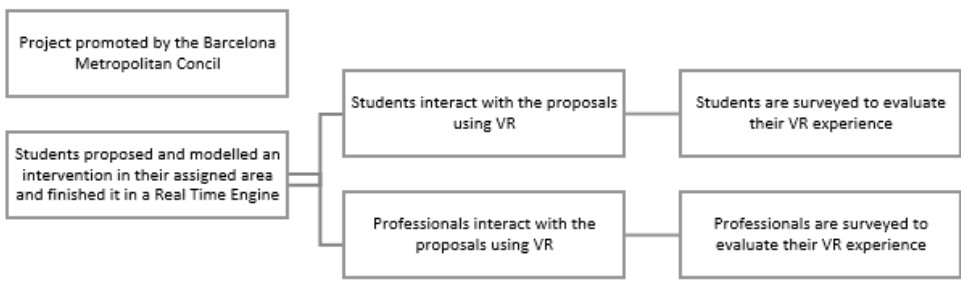

**Figure 5.** Scheme of the process.

We divided the group of users in two. The first group (professionals) was composed of 28 women between 20 and 57 years old (age average, AA: 31.89 and standard deviation, SD: 14.31) and 51 men (AA: 33.86, SD: 14.31, between 20 and 68 years old). In the second group, (architecture students), there were 17 women (AA: 20.76, SD: 1.56, between 19 and 24 years old) and 14 men (AA: 21.14, SD: 2.77, between 19 and 25 years old). We used a Likert scale for the design of the survey for users to evaluate the statement from 1 to 5 based on their level of agreement. The survey has 10 statements about different aspects related to the effectiveness of virtual systems on the design of urban environments

(Table 2). The statements are based on the competences in the curriculum for urbanism of the School of Architecture in Table 1.

**Table 2.** Survey with ten statements about different aspects related to the effectiveness of virtual systems on the design of urban environments, using the Likert scale.

| | | | Disagree -> Agree | | | |
|---|---|---|---|---|---|---|
| **The interactive virtual reality (VR) system helps:** | | 1 | 2 | 3 | 4 | 5 |
| 1. | Easily identify the needs and requirements of the human scale | | | | | |
| 2. | Understand the relationship between people and the natural or artificial environment and objects | | | | | |
| 3. | Critically evaluate the result of an urban design and make decisions | | | | | |
| 4. | Transmit problems, solutions and ideas, to a non-specialized and specialized public | | | | | |
| 5. | Apply formal, functional and technical basic principles to the conception and design of urban complexes | | | | | |
| 6. | Understand the relationship between buildings and the spaces between them | | | | | |
| 7. | Analyze location conditions, establish dimensions and relationships of urban spaces | | | | | |
| **Knowing the interactive VR system:** | | 1 | 2 | 3 | 4 | 5 |
| 8. | It would motivate me to change my way of working in the future | | | | | |
| 9. | I would use it to defend the arguments of urban projects | | | | | |
| 10. | I would use it to defend the arguments of architectural projects | | | | | |

## 3. Results

We analyzed the value given to each of the statements described on the Table 2 and decided to compare the users based on their background (professionals and students) and within each group, by gender (female and male) to see if there are any differences, as previous work [44] has proved that there is a significant difference between these factors. In the present study, we only chose users that were related to the architectural field. Separating professionals from students, we then separated them by gender. In both groups (male and female) of the professional group, statement #5 obtained the lowest value in both groups, while statements #8 and #3 shared the next lowest value (Figure 6). However, the highest values were different. Male users valued statements #4, #6 and #1 as the highest (in that order), and female users valued statements #9, #4 and #2 as the highest (in that order), having statement number #4 in common.

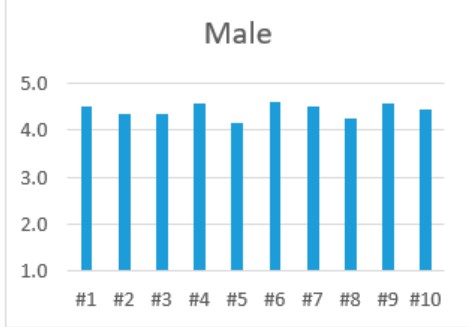
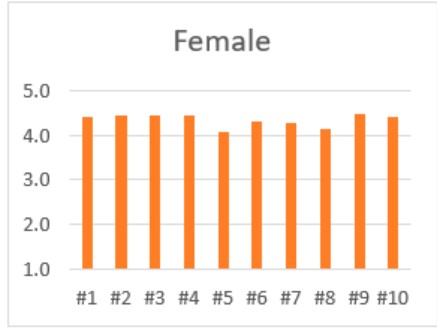

**Figure 6.** Answers of males and females in the professionals' group.

To estimate the probability that profiles by group are significantly similar, we used the Student's *t*-test (Gosset, 1908), using a null hypothesis ($H_0$) that stated that there were no differences in scores between groups. Statistical significance (two-tailed) obtained was $p = 0.2812$, which exceeded the threshold of 0.05, which meant a low probability that the responses based on gender for the professional users were different.

Comparing both groups there were no significant differences. It was barely noticeable that 80% of the statements obtained a higher valuation from the male group, where statements #2 and #3 are the exceptions, were more valued by women (Figure 7). The lowest values were found in the statement #5 with a global average of 4.10 (SD: 0.79) followed by #8 with an average of 4.19 (SD: 0.78), while at the opposite end with averages of 4.51 (SD: 0.66) and 4.49 (SD: 0.77), we found statements #9 and #4).

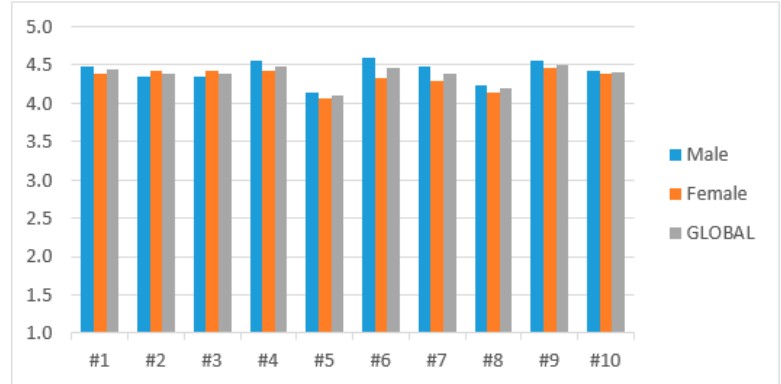

**Figure 7.** Comparison between male, female and global results in the professionals' group.

In the students' group and comparing the global results between male and female students, we did not find a significative difference ($p = 0.8262$) either, see Figure 8.

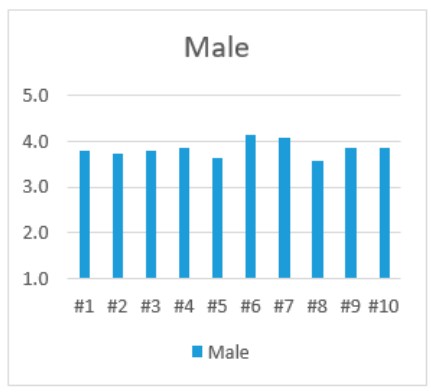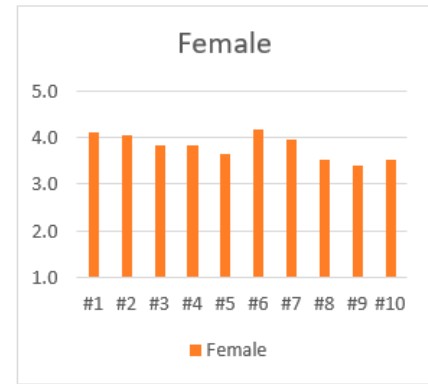

**Figure 8.** Answers of males and females in the students' group.

The lowest values was in statement #8 with a global average of 3.55 (SD: 0.99) followed by #9 with an average of 3.63 (SD: 0.90), while at the opposite end we found statements #6 and #7 with averages of 4.16 (SD: 0.74) and 4.01 (SD: 0.68)), as shown in Figure 9.

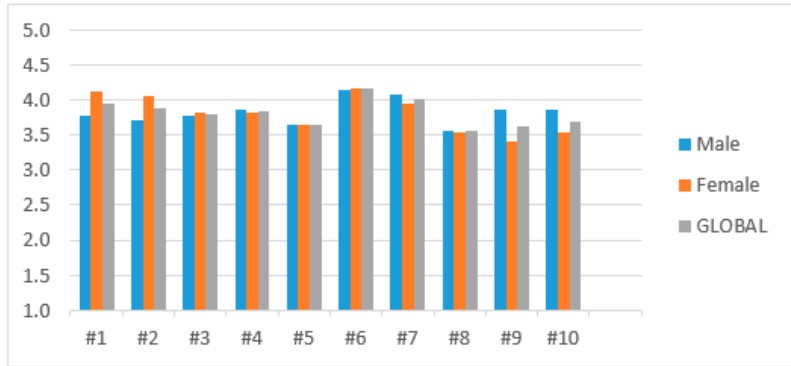

**Figure 9.** Comparison between male, female and global results in the students' group.

When we grouped all professionals together and all students together, the first relevant fact was that the differences between the two groups were statistically significant ($p = 0.000$), which indicated that both groups had clearly differentiated responses as shown in Figure 10.

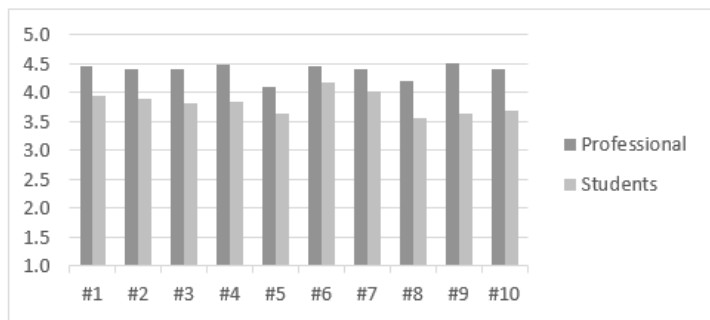

**Figure 10.** Comparison between the professionals and students' group.

The global level for the professionals surveyed, regardless of gender, stood at 4.44 (SD: 0.12), while for students, the overall value was significantly lower (Av: 3.82, SD: 0.19). While the lowest values were grouped into the statements #5 and #8 for both groups, confirming these statements as the perceived ones of less importance and impact at both the educational and professional level. The highest values were distributed differently between groups, being #9 (statement with the greatest difference between groups, of 0.87 points) and #4 for professionals and #6 and #7 for students. The statements were more similar as averages (both with differences that were not statistically significant).

Differences by gender from the same age group were minimal, and there was no gender gap in the use and potential perceived utility of the systems studied, as well as the improvement for compression of the space. However, the age gap was very clear. Professionals saw these systems with great potential and usefulness, while students still did not value them.

## 4. Discussion

After collecting the results, we noticed limited variation in the answers, in terms of gender, having a small difference between them. This aspect was not a variable in this case, in opposition to other case studies [44] where the answers by gender showed a significant difference. Based on this, we can conclude that in the case of studies the gender does not cause any variations. The second comparison that shows slightly differences is the comparison by background. The differences were not very high, but still represented how the stage in career affects the way they agree on the usefulness of a method.

By offering architecture students and professionals new ways to participate in the process of design, they can better visualize and understand physical projects, which allows them to develop both the dimensional and ergonomic relationships between elements, as they see their designs come to

life in real time. It is possible that virtual reality applied in design and local government contexts, might change the way we conceive urban development and planning, even allowing users to think in a 'greener', ecologically-friendly way.

Based on our hypothesis, we can say that gamified strategies for the understanding of three-dimensional space facilitate the acquirement of urban spatial competencies. The use of digital interactive systems in the educational process of urban design courses, helps to improve digital skills in complex representation and allows for the re-evaluation of urban spaces. Regarding the learning aspects from the results and the information gathered on the principles about how people learn, presented in the introduction of this article, we can conclude that:

- Professionals valued the system higher than the students as professionals have more knowledge on the field and know what to do, and what they are connecting is only the how to do it, which is the with the VR system. Meanwhile, students are learning both things, the what and the how at the same time, rather than connecting both aspects with previous knowledge.
- There were more similarities within the students' group than within the professionals' group. The students that participated in the experience were all at the same level of their academic career, while the variation in the professionals' group was higher, as it depended on their professional specializations inside their fields.
- Professionals valued higher the systems that help them to transmit problems, solutions and ideas to both, the non-specialized and specialized, public. Students, valued higher the fact that this system helps them to understand the relationships between buildings and the space between them. The values were associated to those that were more used to working in their careers.
- It is evident that this tool, in general, was very well evaluated because of the interaction it provided to users. This was also found in previous tests [26,44].

Regarding the urban design process, the quantitative analysis reveals that both groups highly value the fact that the interactive VR system helps them easily identify needs and requirements of citizens, a scenario both groups have to work with during the first stage of any urban design process. It is confirmed, through the outcomes, that the use of VR in the design of urban environments improves spatial perception and urban competences and can be used as a method for educational purposes to help in the design process and its representation.

## 5. Conclusions and Future Work

Through this research, it was shown that teaching methodologies can be successfully tackled by using methodologies that adjust to the profile of the student, and just an important, methodologies that adjust with what is used in the professional field. The use of virtual reality in the educational process of urban design courses was demonstrated to aid in the acquisition of urban design competences. This not only revealed the practicality of the system, but also the potential in the academic improvement of the student. Still, a feature to be evaluated in the future is the fact that the participants gave a lower value to the statement that they would use it again in the future. Although the profile of current students is that of a user familiar with the use of technologies to communicate and represent ideas, there is still a gap between the potential of ICTs incorporation in classrooms and its actual implementation in the workforce. It is necessary to change the way these tools are introduced and explained in educational institutions, in order to reduce the gap between the educational sector and the professional sector. The latter seems to be more prepared to incorporate all kinds of technologies, interaction, gamification and different strategies. Our future research will focus on integrating this as a method in the teaching of urban design processes.

**Author Contributions:** Conceptualization, M.V.S.-S.; methodology, R.T.-K., D.F. and J.F.-S.; software, M.V.S.-S.; validation, M.V.S.-S., R.T.-K., and J.F.-S.; formal analysis, M.V.S.-S., R.D.F., and J.F.-S.; investigation, M.V.S.-S.; resources, R.T.-K., and D.F.; data curation, M.V.S.-S., and D.F.; writing—original draft preparation, M.V.S.-S.;

writing—review and editing, M.V.S.-S., R.T.-K., and J.F.-S.; visualization, M.V.S.-S., and R.T.-K.; supervision, R.T.-K., D.F., and J.F.-S.; project administration, D.F.; funding acquisition, D.F.

**Funding:** This research was funded by the National Program of Research, Development and Innovation aimed to the Society Challenges with the references BIA2016-77464-C2-1-R & BIA2016-77464-C2-2-R.

**Conflicts of Interest:** The authors declare no conflicts of interest.

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
