# Peer review of "Methodologies of Learning Served by Virtual Reality: A Case Study in Urban Interventions"

_applsci, doi:10.3390/app9235161_

Round 1
Reviewer 1 Report
Basically, this paper is well-structured, and the authors have listed the benefit of using VR technology in urban environment in terms of education; also, they have provided experimental results based on designing a survey. However, there are still some recommendations to make the quality of this paper better. First, as the title implies “urban,” what are the differences between urban and rural for the use of VR technology in terms of education? Second, it is advised to highlight your designed mechanism, using flowcharts, pseudo codes, or figures to make readers better understand your contributions. Finally, as for submitting to the journal of Applied Sciences, some contributions on algorithms written in formulas or equations are suggested. The section “Conclusion and Future Work” is missing! The authors should have this section as the final one to conclude the content of the paper like most of academic manuscripts.
Author Response
Dear reviewer, we thank you for your comments, suggestions and proposed changes. We have improved our paper according with your review. Thanks a lot for your work.
First, as the title implies “urban,” what are the differences between urban and rural for the use of VR technology in terms of education?
ANSWER: Urban areas are plan settlements while rural areas develop randomly. Our case study is based on settlements were the planning design is applied.
Second, it is advised to highlight your designed mechanism, using flowcharts, pseudo codes, or figures to make readers better understand your contributions.
ANSWER: A procedure flow chart is added on Figure 5. Also other image as Figure 1 was added to complement the visualization of the methodology used.
The section “Conclusion and Future Work” is missing! The authors should have this section as the final one to conclude the content of the paper like most of academic manuscripts.
ANSWER: The section “Conclusion and Future Work” has been added.
Reviewer 2 Report
General Comments:
I would encourage the authors to throughly proof-read their work before submitting it for peer-review (i.e., Highlighted content, poor grammar, incomplete sentences). I would strongly encourage the use of a professional proof-reading and English language editing service.
While the topic under study might be useful, this paper has an unusual format and language for a journal article. For instance, the introduction and results sections have major claims without citations. I would encourage the authors to search for a well cited article utilizing a similar research methodology to fix issues in this paper.
Specific comments:
[11] Describing AR/VR as an …educational method… would be too narrow. I would encourage you to use a broader definition from literature.
[13] The work ‘ student profile ‘ is used very often in your paper. It not clear what you mean by that. Could you reword it?
[36] though -> thought?
[44] remove apostrophe in professor’s
[47] [67] [72] [75] [92] … throughout the article. Check the sentence structure.
[95] citation needed for the claim about current students’ different learning styles. Educational Tech research has disproved this claim multiple times.
[111] here and throughout the article. Remove highlighting
[112 -114] The intent of this sentence is not clear.
[121-123] citation needed for this claim
[126-127] citation needed for this claim. Has it been observed in this or any previous studies?
[137] ‘accessible’ term is most often used in the context of assistive tech for people with disabilities. Might be useful to clarify this.
[147] VR headset is more appropriate than ‘glasses’
[151-152] The statement about lighting is not clear.
[156-157] Check sentence structure.
[158] We emphasis to?
[159] The meaning of ‘serious games’ is not clear
[163] user
[165] check sentence structure
[177] examples?
[279-282] This is a speculation. This claim is beyond the scope of what was done in this study and the authors should refrain from making these claims without sufficient evidence.
[295] Unsubstantiated claim. The cause for this is not yet clear and is beyond the scope of this study.
Author Response
Dear reviewer, thank you for your review, comments, and suggestions. We have improved our paper according with your review. After every review, we have commented the changes done. We hope the new version will be satisfactory for you. Thanks a lot for your work.
I would encourage the authors to throughly proof-read their work before submitting it for peer-review (i.e., Highlighted content, poor grammar, incomplete sentences). I would strongly encourage the use of a professional proof-reading and English language editing service.
ANSWER: We have proof-read our work. We hope the new version will be satisfactory for you.
While the topic under study might be useful, this paper has an unusual format and language for a journal article. For instance, the introduction and results sections have major claims without citations. I would encourage the authors to search for a well cited article utilizing a similar research methodology to fix issues in this paper.
ANSWER: The language and the format was reviewed and fixed.
Specific comments:
[11] Describing AR/VR as an …educational method… would be too narrow. I would encourage you to use a broader definition from literature.
ANSWER: We changed it to “A computer-simulated reality and the human-machine interactions generated by computer technology and wearables, is an educational…” The referenced definition from literature is describe in the article. In the abstract is only is summarized.
[13] The work ‘ student profile ‘ is used very often in your paper. It not clear what you mean by that. Could you reword it?
ANSWER: We leave the sentence without the word “profile” as by saying “current student” was enough to give the description.
[36] though -> thought?
ANSWER: Fixed.
[44] remove apostrophe in professor’s
ANSWER: Removed.
[47] [67] [72] [75] [92] … throughout the article. Check the sentence structure.
ANSWER: Checked.
[95] citation needed for the claim about current students’ different learning styles. Educational Tech research has disproved this claim multiple times.
ANSWER: Fixed.
[111] here and throughout the article. Remove highlighting
ANSWER: Removed. The highlighting was part of the answers to the first round reviewers.
[112 -114] The intent of this sentence is not clear.
ANSWER: Fixed.
[121-123] citation needed for this claim
ANSWER: Citation added.
[126-127] citation needed for this claim. Has it been observed in this or any previous studies?
ANSWER: Changed.
[137] ‘accessible’ term is most often used in the context of assistive tech for people with disabilities. Might be useful to clarify this.
ANSWER: Changed.
[147] VR headset is more appropriate than ‘glasses’
ANSWER: Changed.
[151-152] The statement about lighting is not clear.
ANSWER: More detailed description in the paragraph has been added.
[156-157] Check sentence structure.
ANSWER: Checked.
[158] We emphasis to?
ANSWER: Changed.
[159] The meaning of ‘serious games’ is not clear
ANSWER: Fixed.
[163] user
ANSWER: Changed.
[165] check sentence structure
ANSWER: Checked.
[177] examples?
ANSWER: Changed.
[279-282] This is a speculation. This claim is beyond the scope of what was done in this study and the authors should refrain from making these claims without sufficient evidence.
ANSWER: Changed.
[295] Unsubstantiated claim. The cause for this is not yet clear and is beyond the scope of this study.
ANSWER: Changed.
Round 2
Reviewer 2 Report
The authors have sufficiently addressed my concerns. Please fix minor formatting issues (e.g., figure captions in the same page as figures etc.)
This manuscript is a resubmission of an earlier submission. The following is a list of the peer review reports and author responses from that submission.
Round 1
Reviewer 1 Report
Please label the axes in all your Figures. Some charts are not clear.
The statistical analysis needs work. The authors don't even report the standard deviations of their samples. The conclusions are based on directly comparing the average score for each question. I recommend adding a formal test, t-test for example, to determine if there is a significant difference between the means of the two groups and provide stronger evidence for their conclusions.
Reviewer 2 Report
The paper is focused on the possible utilization of VR with education purposes in the field of architecture and urban design.
Given the quite obvious association between a field focused on the relationship between human beings and physical space on the one hand and a technology like the VR allowing to replicate/simulate physical spaces within explorable synthetic world on the other hand, I would expect some kind of specificity from as study of such nature. On the contrary, the paper remains always very superficial in all its aspects (state-of-the-art, bibliography, definitions, description of the experience, analysis of results, discussion), without insights, technical specificity, deep justifications, true lessons learned.
For example, the research question, from the introduction, goes: “The objective in this paper is to assess how the new ICTs are integrated into the educational processes, enhancing flexibility, motivation and pertinence to the current working demands”, so general to be applicable to almost any context of use in the education field in the past 20 years.
In the conclusion authors say “the use of VR on the design of urban environments improve spatial perception and urban competences, because of the immersive visual technology experience. The application of this system can be used as a method for educational purposes to help in the design process and for its representation.”, which was very much expected, I believe. Furthermore, this too general conclusion comes from a discussion in which authors seem to validate education principles already consolidated in the literature rather than the VR-based system.
Actually, it is NOT EVEN CLEAR what system users did experience: at the beginning of Section 2, where “Methods and Technologies” are presented, authors say that the general purpose of the project is to virtually recreate urban areas of the city of Barcelona, presenting this goal AS STILL TO COME (“This virtual three-dimensional scenario WILL BE an accessible environment”, “the virtual gaming application … WILL ENHANCE … “, “We WILL EMPHASIS ….”); later on the paper says “with RV glasses, the participants experimented and shaped the urban public space”, without any other detail and without any indication of tasks/activities performed by the users during the experiment.
Other flaws:
· The state-of-the-art on Enhanced Technologies Learning (in Section 1.1) is focused almost only on previous works of one of the authors (8 references out of 11), with almost no comparison with related work of other researchers in the field
· Section 1.2 on “Mixed Reality as an educational method” presents VR, augmented reality and mixed reality in a too informal way. It also use a wrong citation (bib item 35, which is a paper on mixed method research and not on mixed reality!!) ignoring - among others - the seminal paper of Milgram&Kishino on mixed reality, and the whole literature on immersivity, sense of presence, relation between body movement and engagement in VR, the role of active exploration etc. etc
· Also data gathering techniques in Section 2.1 are discussed too informally
· The comparison between male and female users in the analysis of results is not enough motivated. Actually, the scale of the chart in Table 6 does not allow to understand if differences between the two groups are really significant and no precise figures are given.
Last but not least, the English need a thorough review.
Reviewer 3 Report
Aside from being a very interesting application, the written form in my opinion is poor, full of errors, imprecisions and unclear sentence. I believe it needs a thorough review.
Some examples:
L. 16 This paper LINKS ?
L. 47 It can be GATHERED
L. 72 The greater is the student engaged
L.118-120: almost unreadable
L.145 dashes
L.174 achieve to gather
L.175 statistically differences
L.176 intended to quantity / QUANTIFY?
L.214 graphic obtain
L.288 We can be said
...and many more.
The paper is also quite repetitive and inflated around the same few concepts.
Section 3: I would add information about what each referred statement is about, otherwise the whole section gets pretty unreadable.
L.112 References are missing here, in my opinion.
Reviewer 4 Report
In this paper, the authors link Virtual Reality as a resource taken into account in the education of courses that work with the design of the urban space. The work is presented in a very comprehensive manner. The research methods are properly discussed. The results discussed in this paper prove that VR helps to expand the digital abilities in complex representation and allow evaluation and decision-making in the processes of urban design spaces. The work is innovative and well-presented with adequate details and discussions. I highly recommend this paper for publication. However, prior to publishing I recommend checking the numbering of figures - it seems figure 3 comes before figure 1.